# High-Mobility Group Box 1 Inhibitor BoxA Alleviates Neuroinflammation-Induced Retinal Ganglion Cell Damage in Traumatic Optic Neuropathy

**DOI:** 10.3390/ijms23126715

**Published:** 2022-06-16

**Authors:** Jingyi Peng, Jiayi Jin, Wenru Su, Wanwen Shao, Weihua Li, Zhiquan Li, Huan Yu, Yongxin Zheng, Liuxueying Zhong

**Affiliations:** State Key Laboratory of Ophthalmology, Zhongshan Ophthalmic Center, Sun Yat-sen University, Guangzhou 510060, China; pengjy7@mail2.sysu.edu.cn (J.P.); jinjiayi@gzzoc.com (J.J.); suwr3@mail.sysu.edu.cn (W.S.); shaowanwen@gzzoc.com (W.S.); liweihua@gzzoc.com (W.L.); lizhiquan@gzzoc.com (Z.L.); yuhuan@gzzoc.com (H.Y.); fdhqs@126.com (Y.Z.)

**Keywords:** TON, ONC, neuroinflammation, RGCs, HMGB1, BoxA, NF-kB, inflammasome

## Abstract

Traumatic optic neuropathy (TON) is a significant cause of vision loss and irreversible blindness worldwide. It is defined as retinal ganglion cell death and axon degeneration caused by injury. Optic nerve crush (ONC), a well-validated model of TON, activates retinal microglia and initiates neuroinflammation. High-mobility group box 1 (HMGB1), a non-histone chromosomal binding protein in the nucleus of eukaryotic cells, is an important inducer of microglial activation and pro-inflammatory cytokine release. The purpose of this study was to examine the protective effects and mechanism of the HMGB1 inhibitor BoxA to neuroinflammation-induced retinal ganglion cells (RGCs) damage in traumatic optic neuropathy. For that purpose, an optic nerve crush model was established in C57BL/6J mice at 10–12 weeks. Model mice received an intravitreal injection of PBS and the HMGB1 inhibitor BoxA. Our data demonstrated that HMGB1 expression increased after optic nerve crush. Retinal ganglion cell function and morphology were damaged, and retinal ganglion cell numbers were reduced after optic nerve crush. Intravitreal injection of BoxA after ONC can alleviate damage. Furthermore, BoxA reduced microglial activation and expression levels of nuclear factor κB (NF-kB), nucleotide-binding domain, leucine-rich repeat containing protein 3 (NLRP3), and apoptosis-associated speck-like protein containing a CARD (ASC) in experimental ONC mice. In summary, HMGB1 mediates NLRP3 inflammasome via NF-kB to participate in retinal inflammatory injury after ONC. Thus, intravitreal injection of BoxA has potential therapeutic benefits for the effective treatment of RGC death to prevent TON.

## 1. Introduction

Traumatic optic neuropathy (TON), defined as retinal ganglion cell (RGC) death and axon degeneration caused by injury [1,2,3,4], is one of the most important causes of severe post-traumatic vision loss [5]. Common causes of TON include car accidents, sporting injuries, and assaults [6,7]. However, there are no effective treatments [5,8,9]. TON appears to involve two phases of injury [8,10], an acute primary injury phase caused directly by the trauma and a delayed secondary injury phase of uncertain etiology but potentially involving neuroinflammation [11].

Optic nerve crush (ONC) is a well-validated model of TON [12]. Researchers found that ONC activates retinal microglia and the recruitment of infiltrating macrophages into the retina [13]. Microglia are the resident immune cells of the central nervous system and the primary initiators of neuroinflammation [14]. Pathological stimuli, such as infection and mechanical tissue damage, transform microglia to an activated state characterized by accelerated proliferation and the release of cytokines that amplify the inflammatory response [15,16]. High-mobility group box 1 (HMGB1), a non-histone chromosomal binding protein in the nucleus of eukaryotic cells, is an important inducer of microglial activation and pro-inflammatory cytokine release [17]. HMGB1 is a critical player in retinal inflammation [18,19], indicating that HMGB1 may trigger neuroinflammation and, therefore, may contribute to TON. BoxA [20,21] is one of the characteristic DNA binding domains of the HMGB1 protein. It consists of 89 amino acids and has a calculated molecular mass of approximately 10.4 kDa. Researchers found that BoxA can inhibit inflammation by binding unproductively to RAGE and competing with functional HMGB1 [22,23]. The inhibition of HMGB1 also reduced neuronal cell loss in the mice after ischemia reperfusion [24].

Therefore, we speculated that HMGB1 contributes to TON-associated retinal inflammation. Specifically, we hypothesized that optical nerve crush induces the passive release of HMGB1. BoxA can inhibit the activation of inflammasome, the death of retinal ganglion cells, and retinal function degeneration. To investigate the effects of HMGB1 on retinal inflammation associated with TON, we compared the effects of ONC in wild-type C57BL/6J mice injected with Phosphate Buffered Saline (PBS) and HMGB1 inhibitor BoxA into the vitreous chamber.

## 2. Results

### 2.1. HMGB1 Is Released in Retina after ONC

To determine the release of HMGB1 after ONC, simple western immunoblots were performed in uninjured controls at 24, 48, and 72 h after experimental ONC. Retinal tissue was collected, and simple western immunoblots were used to detect the expression of HMGB1 (Figure 1a). A simple western immunoblots detection of the whole retina proved that the expression level of HMGB1 in the ONC group significantly increased 48 h after ONC compared with intact controls (*p* < 0.05). This indicates that HMGB1 expression is increased after ONC, and HMGB1 is involved in the inflammatory response in TON.

### 2.2. Optimizing the Dosage of BoxA for Intravitreal Injection

We next explored the optimal dosage of BoxA (ranging from 0.5 to 2 µg/µL) following intravitreal injection post ONC (Figure 2b). As a simple index of RGCs function, we used the N2 wave amplitude of Pattern Electroretinogram (PERG) [25]. Our results show that, following the injection of 1 µg/µL, the average N2 wave amplitude was higher than that of the PBS group (*p* < 0.05). Moreover, it was greater than that of the groups receiving 0.5 µg/µL or 2 µg/µL. Therefore, for the remaining experiments, we applied BoxA at a dosage of 1 µg/µL.

### 2.3. Effects of BoxA in RGCs Function after ONC

To investigate whether HMGB1 participates in RGC function damage after ONC, we performed P-ERG and STR recordings to test inner retinal function [25,26]. In the control uninjured group, retinal neurons responded well to pattern and light stimulation. In the ONC + PBS group, due to retinal damage, the N2 wave of P-ERG and pSTR amplitude values significantly decreased (Figure 2). In contrast, intravitreal injection of the HMGB1 inhibitor BoxA enhanced the amplitude values in the pSTR and N2 wave of PERG (*p* < 0.001, *p* < 0.05). In the pSTR, the amplitude was significantly inhibited in ONC + PBS (*p* < 0.0001) (Figure 2 and Table 1). Following BoxA intravitreal injection, pSTR amplitudes were significantly increased compared to the ONC + PBS group (*p* < 0.001). In the P-ERG response, the amplitude of the N2 wave in ONC + BoxA groups was significantly higher than the ONC + PBS group (*p* < 0.0001). The N2 wave of P-ERG and pSTR amplitude values was normalized to those of the contralateral control eye and expressed as a percentage (Figure 2). A rapid loss was observed on days 6 or 7. Six to seven and thirteen to fourteen days after injury, the amplitude of the ONC + BoxA group was higher than that of the ONC + PBS group; an obvious decreasing trend remained evident, indicating that BoxA could delay the damage of RGCs function, but cannot reverse this damage. Hence, our ERG results suggest that HMGB1 participates in retinal function damage, especially RGCs damage, in experimental ONC mice.

### 2.4. Inhibition of HMGB1 Protects Retinal Structure and Reduces RGCs Apoptosis in Experimental ONC

Improvements in P-ERG and pSTR amplitudes after intravitreal injection of BoxA suggest that RGCs damage was inhibited in experimental ONC mice. Thus, we further examined retinal morphology by Optical Coherence Tomography (OCT). We measured the thickness of the inner retina [27] as a morphological indicator of the status of the RGCs on days 7 and 14 after ONC. Figure 3 shows a representative OCT image obtained on day 14. In the ONC + PBS group, the combined retinal thickness of the nerve fiber layer, ganglion cell layer, and inner plexiform layer was significantly reduced (*p* < 0.0001). Compared to the PBS group, the HMGB1 inhibitor BoxA can inhibit the thinning of retinal layers (Figure 3 and extended Table 1). Thickness was normalized to those of the contralateral control eye and expressed as a percentage. On day 7, there was no statistically significant difference in retinal thickness among the two experimental groups (Figure 3). On day 14, the inner retinal thickness of the BoxA group was significantly higher than that of the PBS group (*p* < 0.0001). The retinal thickness in both experimental groups showed a significant downward trend with time. Then, to understand the survival rate of ganglion cells, we performed RBPMS staining; in the ONC + PBS group, the ganglion cells were obviously apoptotic (*p* < 0.001). Intravitreal injection of BoxA after ONC significantly alleviated RGCs apoptosis (*p* < 0.001) (Figure 4).

### 2.5. HMGB1 Inhibitor BoxA Inhibits the Activation of Retinal Microglia

Through the previous experiments, we found that HMGB1 was involved in RGC damage functionally and structurally after ONC. Microglia are involved in the inflammatory response after nerve injury [14]. Therefore, we speculated whether the HMGB1 inhibitor BoxA could inhibit the activation of microglia after ONC. In most cases, ionized calcium-binding adapter molecule 1 (Iba1) can be used to differentiate microglia from other cells [13]. In the ONC + PBS group, Iba1 (+) cells were mainly distributed in the ganglion cell layer, inner plexiform layer, inner nuclear layer, and outer plexiform layer (*p* < 0.05). ONC increased the number of Iba1 (+) cells (Figure 5). On the contrary, after intravitreal injection of BoxA, the number of Iba1 (+) cells was significantly reduced after ONC (*p* < 0.05). Furthermore, western results showed that BoxA treatment significantly prevented Iba1 protein expression on the retina after ONC (*p* < 0.05). These results indicated that ONC promotes the activation of microglia. In addition, the HMGB1 inhibitor BoxA inhibits the activation of microglia (*p* < 0.05). These results demonstrate that HMGB1 is an essential mediator of microglia cell activation and retinal inflammation in experimental ONC.

### 2.6. HMGB1 Mediates NLRP3 Inflammasome, NF-kB after ONC

Chi et al. [28] found that HMGB1 promotes the activation of NLRP3 inflammasome via the NF-kB pathway to mediate retinal ischemic damage and RGC death. NF-κB is an evolutionarily conserved transcription factor involved in a multitude of biological responses [29]. To identify the molecular mechanism through which HMGB1 regulates inflammatory activation during TON, we examined the protein expression level of NLRP3, ASC, and NF-kB, and found that these were increased in the ONC + PBS group and modestly reduced by intravitreal BoxA injection (*p* < 0.05) (Figure 6). These results indicate that HMGB1 may activate inflammatory responses via the NLRP3 inflammasome and NF-kB. These results suggest that inhibition of HMGB1 after ONC can significantly suppress the expression of NF-kB, NLRP3, and ASC, thereby attenuating inflammatory RGCs damage.

## 3. Discussion

This study confirmed that HMGB1 was involved in retinal inflammatory damage after ONC, mainly in ganglion cells, structurally and functionally. It was further elaborated that HMGB1 was involved in inflammatory injury by activating microglia and regulating the NLRP3 inflammasomes and NF-kB to mediate damage.

### 3.1. Protection of RGCs by BOXA in Traumatic Optic Neuropathy

P-ERG and pSTR were used to assess the function of the RGC function. P-ERG plays an important role in the early assessment of RGC function in glaucoma and optic neuropathies [30]. The positive scotopic threshold response (pSTR) elicits very dim stimuli and is driven by the inner retina, including RGCs [31,32]. In the mouse ONC + PBS model, we found that the amplitude of the N2 wave of P-ERG and pSTR decreased, indicating that the inner retina, especially retinal ganglion cells, was damaged after ONC. This phenomenon is also consistent with previous findings [27,31,33]. With an appropriate dose of 1 µg/µL, our results showed that after intravitreal injection of BoxA, the amplitudes of N2 wave and pSTR waves were significantly higher than those in the ONC + PBS group, indicating that inhibiting HMGB1 could reduce the damage of retinal electrophysiological function caused by TON. In the experimental groups, the amplitudes of N2 wave and pSTR were lower than the control uninjured group. Researchers also found decreased ganglion-cell-associated ERG amplitudes after optic nerve crush in mice [31]. Comparing the two experimental groups on 7 and 14 days, the amplitudes of the ONC + BoxA group were significantly higher than those of the PBS group; however, with the progress of time, both groups showed a downward trend, indicating that BoxA can delay the damage of RGCs, but this damage cannot be reversed. These results may be associated with the damage of nerve axons and the abnormal axoplasmic flow that is subsequently caused [34].

To understand the damage to morphology, we observed the structure of the retina by OCT, which can provide images of retinal sections and be used to analyze the retinal structure. It was found that the combined thickness of the NFL, GCL, and IPL was significantly reduced in the TON + PBS group, and BoxA could inhibit this damage. However, we also found that although the electrophysiology was significantly reduced on day 7, there was no significant difference in retinal thickness among the groups. Researchers also found that in patients with traumatic optic neuropathy, the electrophysiology, which reflects RGC function, deteriorates before significant loss of retinal nerve fiber layer thickness [35]. Yukita M et al. also found the same phenomenon in the study of the structural and functional changes of RGC after mouse ONC [27].

### 3.2. Possible Mechanisms of BoxA Mediates the Neuroinflammation-Induced Damage

In this study, the activation of microglia could be observed after TON. At the same time, the HMGB1 inhibitor BoxA inhibits the activation of microglia. Pathological stimuli, such as infection and mechanical tissue damage, transform microglia to an activated state characterized by accelerated proliferation, the release of cytokines that amplify the inflammatory response, and phagocytic activity [15,16]. Collectively, these findings strongly implicate HMGB1 signaling in TON. Based on these findings from mouse models, we propose that HMGB1 is a mediator of retinal inflammation and retinal structural and functional damage in TON.

To further reveal the mechanism by which HMGB1 regulates inflammatory retinal damage, we examined the expression of NLRP3, ASC, and NF-kB. NF-κB activity was essential for immune responses, and the activation of NF-κB signaling pathways was largely associated with inflammatory diseases [36]. The inflammasome is a multi-protein complex assembled by PRRs, including NLRP1, NLRP3, and NLRC4 [37]. The NLRP3 inflammasome was the first to be discovered [28] and the best characterized of the inflammasomes [38]. ASC is the ligand of NLRP3, and NLRP3 further synthesizes the inflammasome by interacting with ASC. Researchers [33] found that NLRP3 is upregulated in the retina following optic nerve crush injury. Chi et al. [28] found that HMGB1 promotes the activation of NLRP3 inflammasome via the NF-kB pathway to mediate retinal ischemic damage and RGC death. Reports have demonstrated that NLRP3 inflammasome and the pathogenesis of several neuroinflammation diseases had a strong correlation [39]. In our study, the expression level of NLRP3, ASC, and NF-kB was increased by ONC and decreased by the inhibition of HMGB1. These findings indicate that HMGB1 participates in retinal inflammatory damage via NLRP3-inflammasome and the NF-kB pathway in traumatic optic nerve injury and is essential for ensuing secondary inflammatory injury, including the apoptosis of RGCs.

Current treatments for TON include neurotrophic therapy, hormone shock therapy, optic nerve decompression therapy, and combination therapy, but none significantly improve vision [40]. In this study, we demonstrate that the inhibition of HMGB1 effectively reduces the functional and structural inflammatory RGC damage and microglia activation in a mouse model of TON. The HMGB1 inhibitor BoxA can alleviate damage but cannot reverse damage to the function and structure of the retina to a normal state. After ONC, damage to RGCs comes from the inflammatory response and loss of axons [34]. Researchers also found that the neuropeptide NAP participates in ONC [41], indicating that other factors or other pathways need to be studied in the future. Further studies are needed to verify pathogenic mechanisms in human patients and to identify inhibitors feasible for therapeutic use.

## 4. Materials and Methods

### 4.1. Animals

C57BL/6J mice (10−12 weeks of age) were purchased from the Animal Research Center of Zhongshan Ophthalmic Center. All animal care and use protocols adhered to the Association for Research in Vision and Ophthalmology (ARVO) Statement for the use of Animals in Ophthalmic and Vision Research and were approved and monitored by the Committee of Animal Care of the Zhongshan Ophthalmic Center (Permit Number: 2017-012).

### 4.2. Study Design

Animals were randomly assigned to three groups: ONC, ONC + PBS, and ONC + BoxA. Concentrations of BoxA (HM-012, HMGBiotech, Milano, Italy) ranging from 0.5 to 2 ug/ul or equal volumes of PBS were injected into the vitreous after ONC. Berkelaar et al. [42] and Yang Liu et al. [31] found that, 7 days post ONC, the number of cells in the RGC layer was significantly lower in the crushed eye, and the combined thickness of the NFL, GCL, and IPL decreased, gradually becoming stable after 14 days. Therefore, we assessed the functionality and structure of RGCs on days 7 and 14. Meanwhile, retinas were subjected to either immunostaining or simple western immunoblots. The detailed protocol is illustrated in Figure 1.

### 4.3. Establishment of Optic Nerve Crush (ONC) Model

Mice were anesthetized with 1% sodium pentobarbital (50 mg/kg), and 0.5% proparacaine eye drops (s.a. ALCON- COUVREYR n.v., Puurs, Belgium) were applied topically. Under a dissecting microscope, the optic nerve was exposed. A crush injury to the optic nerve was created approximately 1 mm behind the globe by fine-tipped forceps for five seconds [27]. Only the left eye was subjected to ONC, allowing the contralateral, uninjured right eye to serve as a control.

### 4.4. Pattern Electroretinogram (P-ERG)

Pattern electroretinogram (P-ERG) recordings were carried out to assess RGCs function [25,32,43]. Anesthetized mice were transferred onto a heating plate with the mouse superior incisor teeth hooked to a bite bar; the mouse head was gently restrained with head holders. The body was kept at a constant temperature (37.0 °C) with a feedback-controlled heating pad. A small drop of balanced saline was topically applied to prevent corneal dryness during P-ERG recordings. The P-ERG was recorded simultaneously from each eye using a RETI-scan system (Roland Consult, Brandenburg, Germany) through a loop electrode [30]. Visual stimuli consisted of black and white horizontal bars (pattern) generated on LED tablets, which were presented independently to each eye at 10 cm distance (56° vertical × 63° horizontal field; spatial frequency, 0.05 cycles/degree; 98% contrast; 800 cd/m^2^ mean luminance). The recorded electrical signals were averaged. P-ERG waveforms consisted of a positive wave (defined as P1) followed by a slower negative wave with a broad trough (defined as N2), both automatically detected by the software. The N2 signal originates from retinal ganglion cells [25]. Therefore, each waveform was analyzed by measuring N2 amplitude, which is defined as the P-ERG amplitude.

### 4.5. Scotopic Threshold Responses (STR)

STR provides a wave with two components, one positive (pSTR) and one negative (nSTR). We have focused on the pSTR, which originates from the inner retina and is associated with RGC function [31,32,44]. It was discriminated as the positive potential occurring below −3.6 log cds/m^2^ [26]. To measure the pSTR responses of retinal neurons, mice were dark-adapted overnight after P-ERG tests, and their RGC function was assessed with a RETIscan System (Roland Consult, Brandenburg, Germany). Dark-adapted mice were stimulated with flash luminance of −4.0 log cds/m^2^; the pSTR response was recorded via a recording electrode kept in the center of the cornea.

### 4.6. Optical Coherence Tomography (OCT)

The Heldelberg Spectralis^®^ HRA + OCT (SpectralisHRA+OCT+MULTICOLOR, Heidelberg, Germany) was used for data collection. Anesthetized mice were transferred onto a heating plate in front of the camera lens, and retinas were scanned by OCT. The combined thickness of the NFL (Nerve fiber layer), GCL (Ganglion cell layer), and IPL (inner plexiform layer) was collected [27,31,45]. Tests were performed by one skilled technician, overlapping 100 pictures each time.

### 4.7. Immunofluorescence

Eyes were fixed with FAS (Formaldehyde-acetate-ethanol-normal saline) solution (G1109, Servicebio, Wuhan, China) for 24 h at room temperature, embedded in paraffin, and sectioned horizontally into 4-μm thick sections through the optic nerve head along the vertical meridian so that each section included the entire retina from ora serrata in the superior hemisphere to ora serrata in the inferior hemisphere. Sections were dewaxed using environmentally friendly dewaxing solution, heated at high pressure for antigen repair, blocked with goat serum at room temperature for 1 h, and sequentially incubated with primary antibody AIF-1/Iba1 ((E4O4W) XP^®^ Rabbit mAb, 17198T, Cell signaling, Boston, Massachusetts, USA) at 4 °C overnight and secondary antibody (Anti-rabbit IgG (H+L), F(ab’)2 Fragment (Alexa Fluor^®^ 488 Conjugate, 4412S, SIGMA, Darmstadt, Germany) at room temperature for 40 min. Stained sections were sealed with anti-fluorescence quenching agent containing DAPI (ab104139, abcam, Cambridge, UK) for the counterstaining of the nucleus. For each retina, values from 3 images were averaged to provide one data point, and all the data points from the retinas of a group were further averaged to obtain the mean value. The Iba1-positive cells were analyzed by using ImageJ software (v1.51, National Institutes of Health, Bethesda, MD, USA).

### 4.8. Flat-Mounted Retinas and RGC Quantification

Harvested eyeballs were immersed in 4% paraformaldehyde (AMS0001, Aomabio, Guangdong, China) at room temperature for 1 h, followed by removal of the sclera, uveal tissue, cornea, and lens. Isolated retinas were washed twice with phosphate-buffered saline (PBS) containing 0.1% Triton X-100 (04807423, MP Biomedicals, Santa Ana, California, USA) and simultaneously were gently shaken for 15 min, blotted with PBS containing 3% BSA (FA016, GENVIEW, Beijing, China) plus 0.1% Triton X-100 at room temperature for 2 h, then incubated with primary antibody targeting the RGC-specific marker RNA Binding Protein with multiple splicing (RBPMS, ABN1362, Millipore, USA) overnight at 4 °C. Labeled retinas were washed 3 times with 0.1% Triton X-100 in PBS for 15 min on a shaker, incubated with secondary antibody (Anti-rabbit IgG (H+L), F(ab’)2 Fragment (Alexa Fluor^®^ 488 Conjugate, 4412S, SIGMA, Darmstadt, Germany) in the same solution for 1 h at room temperature, and rewashed 3 times with 0.1% triton X-100/PBS for 15 min on a shaker. Stained retinal tissues were then flat mounted on glass slides with anti-fluorescence quenching agent and coverslips. The retinas were examined for RGCs at 1 mm from the optic nerve head to provide the central RGC densities [46]. Four 200× images were obtained from each of the four quadrants. The RGCs numbers were analyzed by using ImageJ software (National Institutes of Health, Bethesda, MD, USA).

### 4.9. Automated Western Immunoblotting

A capillary-based immunoassay was performed using the Size Separation Master Kit (SM-W004, ProteinSimple, San Francisco, CA, USA) with Split Buffer (12–230 kDa) by the Wes (WS3312, ProteinSimple, San Francisco, CA, USA) [47,48,49,50]. Protein expression was measured by chemiluminescence and quantified as area under the curve using the Compass for Simple Western program (Compass for SW 5.0.1, ProteinSimple, San Francisco, CA, USA). Proteins were detected with the following primary antibodies: NF-kB p65 (8242, cell signaling); HMGB1 (D3E5) (6893, Cell signaling, Boston, MA, USA); NLRP3 (D4D8T) (15101S, cell signaling, Boston, MA, USA); ASC/TMS1 (D2W8U) (67924, Cell signaling, Boston, MA, USA); GAPDH (5174, Cell signaling, Boston, MA, USA).

### 4.10. Statistical Analysis

All data are presented as the mean ± SD or a percentage. Treatment group means were compared by one-way ANOVA and Student’s t-test as indicated using GraphPad Prism software (version 8.0, GraphPad Software, Inc., San Diego, CA, USA). The level of significance adopted in the present study was *p* < 0.05.

## 5. Conclusions

In summary, our current study demonstrated that the inhibition of HMGB1 by BoxA effectively reduces inflammatory RGC damage functionally and structurally in a mouse model of TON. The observed decrease in the expression level of NLRP3 inflammasome and NF-kB could be the underlying molecular mechanism. Thus, intravitreal injection of BoxA is a possible effective therapeutic treatment of RGC death to prevent TON.

## Figures and Tables

**Figure 1 ijms-23-06715-f001:**
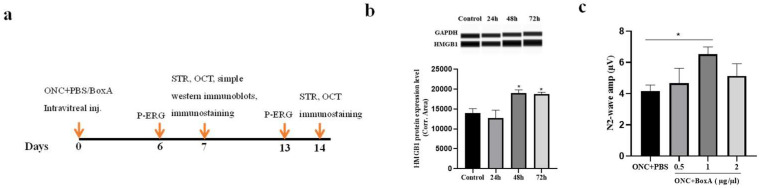
Experimental design. (**a**) In vivo study design. (b) In vivo HMGB1 protein study. (**c**) In vivo BoxA studies. Each pieces of data represents the mean ± SD, *n* = 3. * *p* < 0.05.

**Figure 2 ijms-23-06715-f002:**
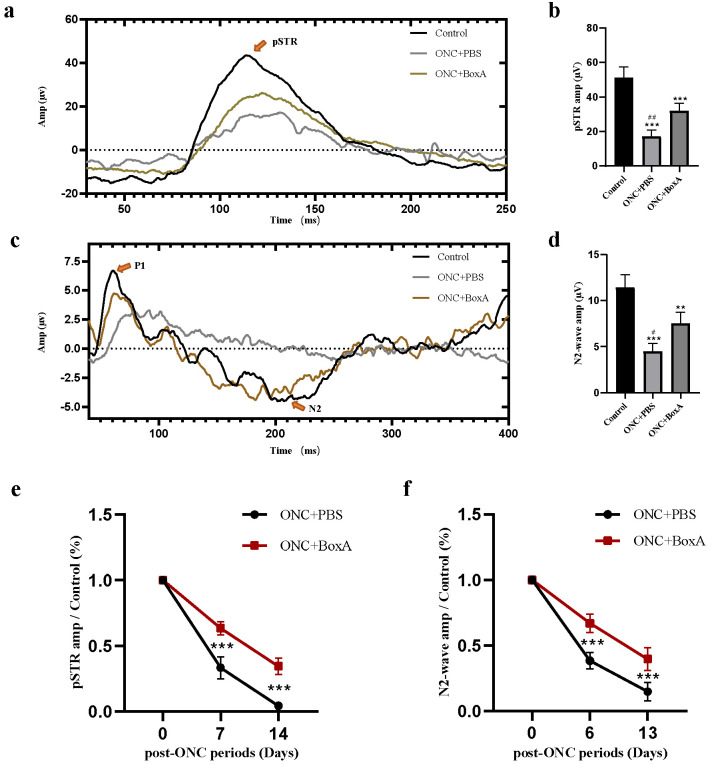
BOXA prevents RGCs function. (**a**) Representative Positive Scotopic Threshold Responses (pSTRs) of the control (black line), TON + PBS group (grey line), and TON + BoxA group (brown line) 7 days post ONC. (**b**) Averaged data of the pSTR amplitudes from recordings. (**c**) Representative Pattern Electroretinogram (PERG) of the control (black line), TON + PBS group (grey line), and TON + BoxA group (brown line) 6 days post ONC. (**d**) Averaged data of the N2 wave amplitudes from recordings. (**e**,**f**) The pSTR and N2 wave amplitudes were normalized to those of the control eyes, expressed as a percentage, and plotted as a function of the post-ONC time course. Each piece of data represents the mean ± SD, *n* = 6. ** *p* < 0.001, *** *p* < 0.0001 vs. control normal eye, # *p* < 0.05, ## *p* < 0.001 vs. TON + BoxA.

**Figure 3 ijms-23-06715-f003:**
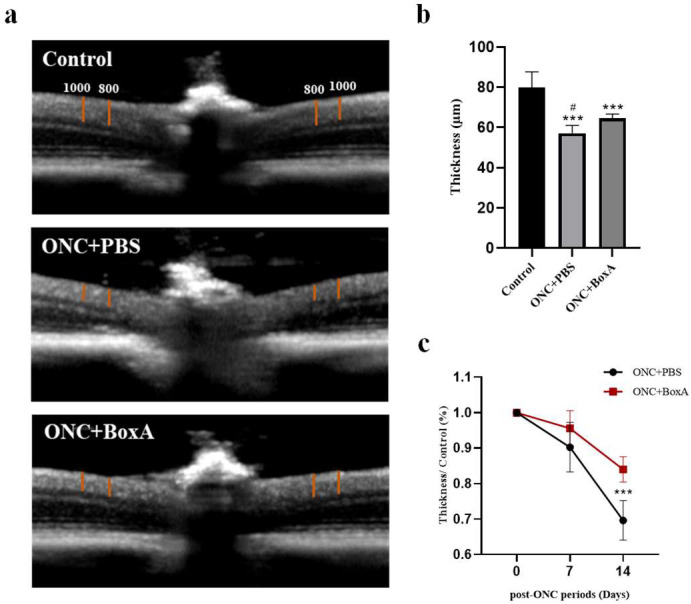
BoxA inhibits the thinning of retinal layers determined by OCT. (**a**) Representative image of the control, TON + PBS, and TON+ BoxA. Two vertical calipers were placed on each side of the optic nerve head, 800 and 1000 µm away from the center of the optic nerve head. The combined thickness (µm) of the NFL, GCL, and IPL was measured. (**b**) Averaged data of retinal thickness from recordings. (**c**) The inner retina thickness was normalized to those of the control eyes, expressed as a percentage, and plotted as a function of the post-ONC time course. Each piece of data represents the mean ± SD, *n* = 6. *** *p* < 0.0001 vs. control normal eye, # *p* < 0.05 vs. TON+ BoxA.

**Figure 4 ijms-23-06715-f004:**
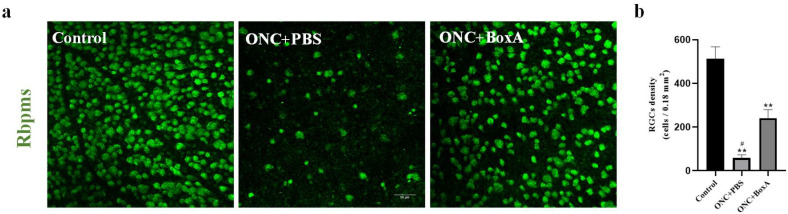
BoxA reduces RGC cell loss following ONC. (**a**) RBPMS staining of the retina. Each figure is representative of 1000 um away from the center of the optic head (428µm × 428µm). Scale bars: 50 µm. (**b**) Cell numbers of RGCs. Each piece of data represents the mean ± SD, *n* = 4. ** *p* < 0.001 vs. control normal eye, # *p* < 0.05 vs. TON+ BoxA.

**Figure 5 ijms-23-06715-f005:**
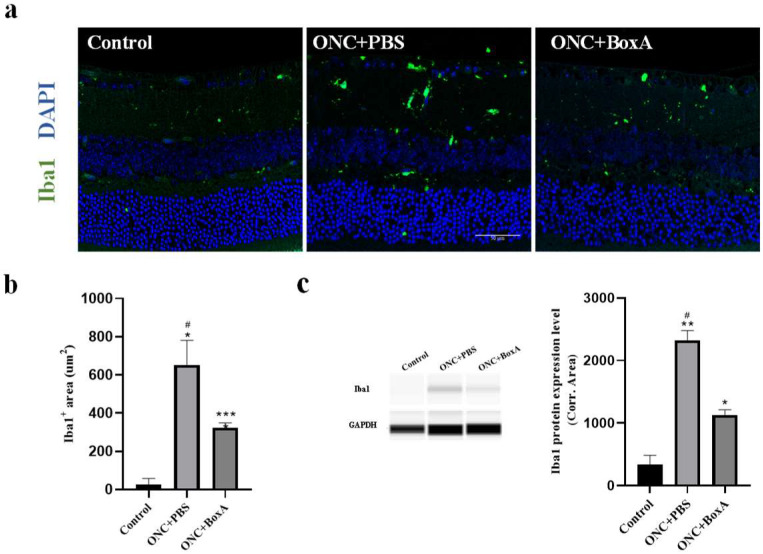
BoxA reduces microglial activation following ONC. (**a**) Retinal slices stained for DAPI (blue) and Iba1 (green). Scale bars: 50 µm. (**b**) Area of Iba1-positive staining per image (250 µm × 250 µm) for each group, *n* = 4. (**c**) The protein level of Iba1 was evaluated. Each piece of data represents the mean ± SD, *n* = 3. * *p* < 0.05, ** *p* < 0.001, *** *p* < 0.0001 vs. control normal eye, # *p* < 0.05 vs. TON+ BoxA.

**Figure 6 ijms-23-06715-f006:**
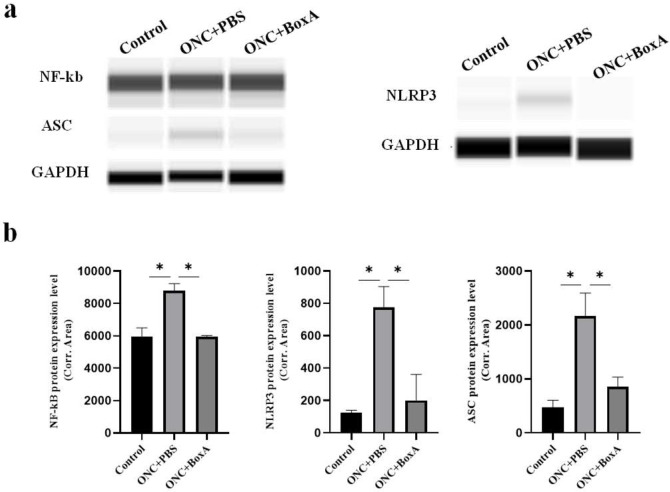
The effects of BoxA on the activation of NLRP3, ASC, and NF-kB. (**a**) The protein level of NLRP3, ASC, and NF-kB in the retina from the control normal group, ONC + PBS group, and ONC+ BoxA group was examined by simple western immunoblots. GAPDH was used to ensure equal loading. (**b**) Analysis of the effects of BoxA on protein expression. Each piece of data represents the mean ± SD, *n* = 3. * *p* < 0.05.

**Table 1 ijms-23-06715-t001:** Mean amplitudes of ERG waves and the combined thickness of the nerve fiber layer (NFL), ganglion cell layer (GCL), and inner plexiform layer (IPL). Each piece of data represents the mean ± SD.

Treatment		Mean Amplitude of ERG Waves ± SD (µv)			Thickness ± SD (µm)
		N2		pSTR	
Control		11.44 ± 1.33		51.07 ± 6.20	79.87 ± 7.46
ONC + PBS	6 d	4.47 ± 0.83	7 d	17.23 ± 3.46	72.95 ± 3.95
	13 d	1.72 ± 0.73	14 d	2.23 ± 0.61	57.14 ± 3.58
ONC + BoxA	6 d	7.54 ± 1.14	7 d	31.88 ± 4.09	73.69 ± 4.91
	13 d	4.50 ± 1.14	14 d	17.08 ± 2.93	64.66 ± 1.85

## Data Availability

The datasets used and/or analyzed during the current study are available from the corresponding authors upon reasonable request.

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
