# Peer review of "High-Mobility Group Box 1 Inhibitor BoxA Alleviates Neuroinflammation-Induced Retinal Ganglion Cell Damage in Traumatic Optic Neuropathy"

_ijms, 2022, doi:10.3390/ijms23126715_

Round 1

Reviewer 1 Report

The manuscript in reference describes the examination of the protective effects and action mechanism of BoxA (an HMGB1 inhibitor) on neuroinflammation-induced RGCs damage in traumatic optic neuropathy. The manuscript is well-written and has relevant information and results that will be interesting for readers. However, some issues should be addressed prior to further consideration.

1.       Detailed scrutiny should be performed throughout the manuscript to revise/correct some grammar and stylistic issues since some passages and sentences are difficult to be followed. An editing service is then recommended.

2.       Page 2, paragraph 2: An expanded explanation about BoxA (For instance, size, structure, variations, etc.) is missing and required for readers. Something similar to HMBG1 protein.

3.       Figures 1,2,3,4,5: Improve the quality and size of these Figures. Some information is difficult to read and observe.

4.       The discussion section can be improved since a clear order is difficult to be followed. It can be even reorganized, possibly by subheadings. The manuscript has numerous significant results that have not been adequately discussed.

5.       Revise in detail the M&M section. Some experimental details are missing to ensure outcome reproducibility. For instance, the brand, model, and grade of reagents, solvents, materials, and instruments must be provided. In addition, the origin of BoxA is missing and it is very important to be informed.

6.       A conclusion section, compiling specific conceptual findings from a mechanistic point of view, is missing and it will be helpful for readers.

Author Response

Point 1: Detailed scrutiny should be performed throughout the manuscript to revise/correct some grammar and stylistic issues since some passages and sentences are difficult to be followed. An editing service is then recommended.

Response 1: Thank you very much. In order to improve the quality of English in the manuscript, we seek professional help from editing services listed at https://www.mdpi.com/authors/english in revising this manuscript, for help with English usage.

Point 2: Page 2, paragraph 2: An expanded explanation about BoxA (For instance, size, structure, variations, etc.) is missing and required for readers. Something similar to HMBG1 protein.

Response 2: Thank you very much. We added expanded explanation about BoxA in page2, paragraph 2. The details are as follows:

“BoxA [20, 21] is one of the characteristic DNA binding domains of the HMGB1 protein. It consists of 89 amino acids and has a calculated molecular mass of approximately 10.4kDa. Researchers found that BoxA can inhibit inflammation by binding unproduc-tively to RAGE and competing with functional HMGB1 [22, 23].”

  1. Yang, H.; Ochani, M.; Li, J.; Qiang, X.; Tanovic, M.; Harris, H. E.; Susarla, S. M.; Ulloa, L.; Wang, H.; DiRaimo, R.; Czura, C. J.; Wang, H.; Roth, J.; Warren, H. S.; Fink, M. P.; Fenton, M. J.; Andersson, U.; Tracey, K. J., Reversing established sepsis with antagonists of endogenous high-mobility group box 1. Proceedings of the National Academy of Sciences of the United States of America 2004, 101, (1), 296-301.
  2. Li, J.; Kokkola, R.; Tabibzadeh, S.; Yang, R.; Ochani, M.; Qiang, X.; Harris, H. E.; Czura, C. J.; Wang, H.; Ulloa, L.; Wang, H.; Warren, H. S.; Moldawer, L. L.; Fink, M. P.; Andersson, U.; Tracey, K. J.; Yang, H., Structural basis for the proinflammatory cytokine activity of high mobility group box 1. Molecular medicine (Cambridge, Mass.) 2003, 9, (1-2), 37-45.
  3. Sitia, G.; Iannacone, M.; Müller, S.; Bianchi, M. E.; Guidotti, L. G., Treatment with HMGB1 inhibitors diminishes CTL-induced liver disease in HBV transgenic mice. Journal of leukocyte biology 2007, 81, (1), 100-7.
  4. De Mori, R.; Straino, S.; Di Carlo, A.; Mangoni, A.; Pompilio, G.; Palumbo, R.; Bianchi, M. E.; Capogrossi, M. C.; Germani, A., Multiple effects of high mobility group box protein 1 in skeletal muscle regeneration. Arteriosclerosis, thrombosis, and vascular biology 2007, 27, (11), 2377-83.

Point 3: Figures 1,2,3,4,5: Improve the quality and size of these Figures. Some information is difficult to read and observe.

Response 3: Thank you very much. We inserted High quality of Figures 1,2,3,4,5,6 in the revised manuscript. Please review the figures in the attachment.

Point 4: The discussion section can be improved since a clear order is difficult to be followed. It can be even reorganized, possibly by subheadings. The manuscript has numerous significant results that have not been adequately discussed.

Response 4: Thank you very much. We cherish this valuable opinion. We have further improved the Discussion section with two subheadings:

3.1. Protection of RGCs by BOXA in traumatic optic neuropathy

3.2. Possible mechanisms of BoxA mediates the neuroinflammation-induced damage

More discussion about results are carried out, specifically in the discussion section of the revised manuscript.

Point 5: Revise in detail the M&M section. Some experimental details are missing to ensure outcome reproducibility. For instance, the brand, model, and grade of reagents, solvents, materials, and instruments must be provided. In addition, the origin of BoxA is missing and it is very important to be informed.

Response 5: Thank you very much. We have improved the brand and product number information of the reagents in M&M section. The details are as follows: BoxA (HM-012, HMGBiotech); 0.5% proparacaine eye drops (Alcon Inc.); 4% paraformaldehyde (AMS0001, Aomabio); Triton X-100 (04807423, MP Biomedicals); 3% BSA (FA016, GENVIEW); Wes (Wes, ProteinSimple).

Point 6: A conclusion section, compiling specific conceptual findings from a mechanistic point of view, is missing and it will be helpful for readers.

Response 6: Thank you very much. We added a conclusion section in page 10, paragraph 2. The details are as follows:

5.Conclusion

In summary, our current study demonstrated that the inhibition of HMGB1 by BoxA effectively reduces inflammatory RGC damage functionally and structurally in a mouse model of TON. The observed decrease in the expression level of NLRP3 inflammasome and NF-kB could be the underlying molecular mechanism. Thus, intravitreal injection of BoxA is a possible effective therapeutic treatment of RGC death to prevent TON.

Reviewer 2 Report

The authors present their manuscript for consideration of publication in International Journal of molecular sciences. This study evaluated the impact of HMGB1 inhibitor BoxA on the traumatic optic neuropathy through the optic nerve crush in mouse.

The paper is well written, however, some minor comments remain:

1- Please, define HMGB1 in title and abstract.

2- In one sentence, explain the key role of HMGB1 in the abstract.

3- Please, change the order of sub-figure by starting with 1c.

4- Could you, please, draw the flow chart of the cascade mechanism of different molecules and inhibitors in the TON inflammation in this paper.

Author Response

Point 1: Please, define HMGB1 in title and abstract.

Response 1: Thank you very much. We have detailed HMGB1 in title and abstract as follows:

  • Title: High-mobility group box 1 inhibitor BoxA alleviates neuroinflammation-induced retinal ganglion cell damage in traumatic optic neuropathy.
  • Abstract: “High-mobility group box 1 (HMGB1), a non-histone chromosomal binding protein in the nucleus of eukaryotic cells, is an important inducer of microglial activation and pro-inflammatory cytokine release.”

Point 2: In one sentence, explain the key role of HMGB1 in the abstract.

Response 2: Thank you very much. We added expanded explanation about HMGB1 in Abstract section. The details are as follows: High-mobility group box 1 (HMGB1), a non-histone chromosomal binding protein in the nucleus of eukaryotic cells, is an important inducer of microglial activation and pro-inflammatory cytokine release.

Point 3: Please, change the order of sub-figure by starting with 1c.

Response 3: Thank you very much. We changed the order of Figure. 1. Please see the attachment.

Point 4: Could you, please, draw the flow chart of the cascade mechanism of different molecules and inhibitors in the TON inflammation in this paper.

Response 4: Thank you very much. A diagram of the cascade mechanism of different molecules and inhibitors in the TON inflammation in this paper is shown in the attachment.
